# Empirical Limitations of the NTK for Understanding Scaling Laws in Deep Learning

**Nikhil Vyas**                                                                  *nikhil@g.harvard.edu*
*Harvard University*

**Yamini Bansal**                                                                 *ybansal@google.com*
*Google DeepMind*[*]

**Preetum Nakkiran**                                                            *preetum@nakkiran.org*
*University of California, San Diego*[†]

**Reviewed on OpenReview:** *https://openreview.net/forum?id=XXXX*

## Abstract

The "Neural Tangent Kernel" (NTK) (Jacot et al., 2018), and its empirical variants have been proposed as a proxy to capture certain behaviors of real neural networks. In this work, we study NTKs through the lens of scaling laws, and demonstrate that they fall short of explaining important aspects of neural network generalization. In particular, we demonstrate realistic settings where finite-width neural networks have significantly better data scaling exponents as compared to their corresponding empirical and infinite NTKs at initialization. This reveals a more fundamental difference between the real networks and NTKs, beyond just a few percentage points of test accuracy. Further, we show that even if the empirical NTK is allowed to be pre-trained on a constant number of samples, the kernel scaling does not catch up to the neural network scaling. Finally, we show that the empirical NTK continues to evolve throughout most of the training, in contrast with prior work which suggests that it stabilizes after a few epochs of training. Altogether, our work establishes concrete limitations of the NTK approach in understanding scaling laws of real networks on natural datasets.

## 1 Introduction

The seminal work of Jacot et al Jacot et al. (2018) introduced the "Neural Tangent Kernel" (NTK) as the limit of neural networks with widths approaching infinity. Since this limit holds provably under certain initializations, and kernels are more amenable to analysis than neural networks, the NTK promises to be a useful reduction to understand deep learning. Thus, it has initiated a rich research program to use the NTK to explain various behaviors of neural networks, such as convergence to global minima (Du et al., 2018; 2019), good generalization performance (Allen-Zhu et al., 2018; Arora et al., 2019a), implicit bias of networks (Tancik et al., 2020) as well as neural scaling laws (Bahri et al., 2021).

In addition to the infinite NTK, the *empirical NTK* — the kernel with features that are gradients of a finite-width neural network— can be a useful object to study, since it is an approximation to both the true neural network and the infinite NTK. This has also been studied extensively as a tool to understand deep learning (Fort et al., 2020; Long, 2021; Paccolat et al., 2021; Ortiz-Jiménez et al., 2021).

In this work, we probe the upper limits of this research program: we want to understand the extent to which understanding NTKs (empirical and infinite) can teach us about the success of neural networks. We

---

[*]Work done while at Harvard University.
[†]Work done while at UCSD; currently at Apple.

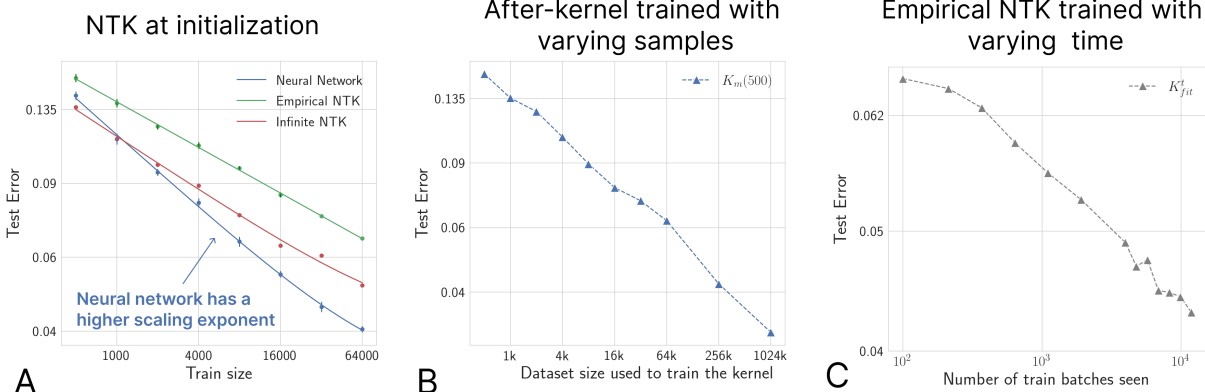

Figure 1: **Summary of results:** (A) *Neural network scales better than NTK at initialization:* We compare the scaling exponent of a neural network, its corresponding infinite and empirical NTK at initialization. Details in Section 3. (B) *After-kernel continues to improve with more training samples:* We train a neural network with $m = \{1K, 2K...1024K\}$ samples, extract the empirical NTK at completion, and use this kernel to fit 500 samples. Details in Section 4. (C) *Empirical NTK improves with constant rate with respect to training time:* We extract the empirical NTK at various times in training and use it to fit the full train dataset. Details in Section 5.

study this question under the lens of scaling (Kaplan et al., 2020; Rosenfeld et al., 2019)—how performance improves as a function of samples and as a function of time— since the scaling is an important "signature" of the mechanisms underlying any learning algorithm. We thus compare the scaling of real networks to the scaling of NTKs in the following ways.

1. *Data scaling of initial kernel (Section 3)*: We show that both the infinite and empirical NTK (at initialization) can have worse data scaling exponents than neural networks, in realistic settings (Figure 1). We find that this is robust to various important hyperparameter changes such as learning rate (in the range used in practice), batchsize and optimization method.

2. *Width scaling of initial kernel (Section 3):* Since neural networks provably converge to the NTK at infinite width, we investigate why the scaling behavior differs at finite width. We show (Figure 2(b), 2(c)) realistic settings where as the width of the neural network increases to very large values, the test performance of the network gets *worse* and approaches the performance of the infinite NTK, unlike existing results in literature which suggest that increasing the width in the over-parameterized regime is always good. This also raises new questions about scaling of neural networks with width, in particular the "variance-limited" neural scaling regimes (Bahri et al., 2021).

3. *Data scaling of after-kernel (Section 4):*We consider the after-kernel (Long, 2021) i.e. empirical NTK extracted after training to completion on a fixed number of samples. We show (Figure 1(B), 4(b)) that the after-kernel continues to improve as we increase the training dataset size. On the other hand, we find (Figure 4(c)) that the scaling exponent of the after-kernel, extracted after training on a fixed number of samples remains *worse* than that of the corresponding neural network.

4. *Time scaling (Section 5):* We show (Figure 1(C), 5(a)) realistic settings where the empirical NTK continues to improve uniformly throughout most of the training. This is in contrast with prior work (Fort et al., 2020; Ortiz-Jiménez et al., 2021; Atanasov et al., 2021; Long, 2021) which suggests that the empirical NTK changes rapidly in the beginning of the training followed by a slowing of this change.

We demonstrate these phenomena occur in certain settings which are based on real, non-synthetic data, and modern architectures (for e.g.: for datasets CIFAR-10 and SVHN and convolutional networks). While we

do not claim that these phenomena manifest for *all possible* datasets and architectures, we believe that our examples highlight important limitations to the use of NTK to understand the test performance of neural networks. Formalizing the set of distributions or architectures for which these phenomenon occur is an important direction for future theoretical research.

### 1.1 Comparison to Prior Work on NTK Generalization

Our main focus is to understand feature learning occurring due to finite width. To do this, we make the following deliberate choices in all of our experiments: a) We use the NTK parameterization, this makes sure that infinite width networks will be equivalent to kernels b) We use the same optimization setup for the neural network, empirical NTK and Infinite NTK, this makes sure that as width tends to infinity all 3 models will have the same limit. We make sure that our comparisons are robust by c) using scaling laws to compare these models and d) doing various hyperparameter ablations (Figure 3).

Below we describe several lines of related works and how our work differs from them.

**Small initialization and representation learning at infinite width.** Infinite widths neural networks in the NTK and standard initialization are equivalent to kernels (Jacot et al., 2018; Yang & Hu, 2021). On the other hand it has been shown (Yang & Hu, 2021; Sirignano & Spiliopoulos, 2019; Nguyen & Pham, 2020; Araújo et al., 2019; Fang et al., 2020) that with small initialization feature learning is possible at infinite width. The feature learning displayed in our experiments is not due to small initialization as we initialize our networks in the NTK parameterization. This was a deliberate choice as we are interested in feature learning occurring due to finite width as this is the kind of feature learning displayed by empirical neural networks (which usually do not have a small initialization).

**Data Scaling for NTKs and neural networks.** Scaling laws have been empirically shown Kaplan et al. (2020); Rosenfeld et al. (2019) for neural networks and have been theoretically proven Bordelon et al. (2020); Canatar et al. (2021); Bordelon & Pehlevan (2022) for NTKs under natural assumptions. Comparison between the scaling laws for neural network and empirical NTKs has been previous looked at by Paccolat et al. (2021) and Ortiz-Jiménez et al. (2021) and both find that neural networks have better scaling than empirical NTK at initialization. Both of these papers do not compare to infinite NTKs which leaves open the possibility that neural networks and infinite width NTKs behave the same wrt their scaling constants.

**Pointwise comparisons of neural networks and corresponding infinite NTKs** has also been studied extensively in the literature (Arora et al., 2019b; Lee et al., 2020; Simon et al., 2021) but the results have been divided. As discussed earlier we focus on comparing scaling laws. We argue that scaling laws, instead of point-wise comparisons, are the appropriate tool to compare neural networks and NTKs. Practically, pointwise comparisons between any two models can be fraught with issues as the ordering can flip depending on dataset size, as well as the specific choice of hyperparameters. On the other hand, scaling exponents have been found to be more robust to the choice of hyperparameters (Bansal et al., 2022; Kaplan et al., 2020). More importantly, the claim that NTK can capture "most" of the performance of the neural network can be subjective, specially when we are comparing small error or loss values. We show that when we look closely at the scaling exponents of these objects instead, we find major differences.

**Theoretical studied effects of finite width with respect to the NTK regime.** Finite width corrections to the NTK theory have been studied by Andreassen & Dyer (2020); Roberts et al. (2021); Bahri et al. (2021). While these results do not need infinite widths they still require much higher than practically used widths particularly for the training sizes used in practice. These papers either consider a) the finite width corrections of empirical NTK or b) they consider the change in NTK but predict that the higher order analogues of empirical NTK remain constant. For a) we show that the empirical NTK is very far from the performance of finite width neural networks. Regarding b), in Appendix D we show that the higher order analogues of empirical NTK change significantly.

**After Kernel and Time Dynamics** We discuss these in detail in Section 4 and 5.

We describe other related works in Appendix G.

## 2 Experimental Methodology

Here we describe the common methodology used in our experiments.

The core object we want to understand is the data-scaling law of real neural networks— that is, what is its asymptotic performance as a function of the number of train samples? Concretely, in this work we restrict to classification problems, where we measure performance in terms of test classification error. For a given classification algorithm, let $L(n)$ be its *learning curve*: its expected test error as a function of number of samples $n$. In practice, many neural networks exhibit power-law decay in their learning curves (Kaplan et al., 2020). In such settings, we have $L(n) \sim \alpha n^{\beta}$ and we are interested primarily in the *scaling exponent* $\beta$, which determines the asymptotic rate of convergence.

**Empirical and Infinite NTK** Let $f(w, x)$ be a neural network with $w$ representing the weights and $x$ an input. By Taylor expansion around $w_0$ we have:

$$f(w, x) = f(w_0, x) + \nabla_w f(w, x)|_{w_0}(w - w_0) + \frac{1}{2}(w - w_0)^T \nabla_w^2 f(w, x)|_{w_0}(w - w_0) + \dots$$

Empirical NTK of the neural network around weights $w_0$ refers to the model $g_1(w, x) = \nabla_w f(w, x)|_{w_0}(w - w_0)$. Note that this is not the same as linearizing the network as we omit the $f(w_0, x)$ term. Empirical NTK is a linear model with respect to the weights $w$. Infinite NTK refers to the limit of the empirical NTK of the network around initial weights as width tends to infinity.

For a given learning problem and given neural network architecture NN, we want to understand its data-scaling law $L_{\text{NN}}(n)$. We consider the infinite NTK of the NN and the empirical NTK of the NN at initialization and their corresponding learning curves, $L_{\text{NTK}}(n)$ and $L_{\text{ENTK}}(n)$. Now we ask: is the scaling-exponent of $L_{\text{NN}}$ always close to the scaling-exponent of either $L_{\text{ENTK}}$ or $L_{\text{NTK}}$, in realistic settings? That is, how well does the NTK approximation capture the generalization of real networks, on natural distributions?

Recall that this question is especially interesting because the three objects involved (Neural Network, NTK, and ENTK) all become provably equivalent in the appropriate width $\to \infty$ limit. Thus, at infinite-width we know their scaling laws must be the equivalent. The question is then, how far are we from this limit in practice? Are the widths used in practice large enough for their scaling-behavior to be captured by the infinite-width limit? To probe these questions, we empirically study scaling laws of these methods on image-classification problems.

**Remark on comparisons.** We intentionally only compare a neural network to *its corresponding NTK*, and not to other kernels. Our motivation not address the question of "can (some) kernel perform as well as as a given neural network?"— indeed, there may be some better kernel to consider than the NTK. However, our goal is to study the specific kernels given by the NTK approximation, in correspondence with real networks.

**Datasets.** We use the following datasets:

1. A 2 class subset (dog, horse) of CIFAR-5m (Nakkiran et al., 2021) dataset, as a binary classification problem, which we denote *CIFAR-5m-bin*. This is a dataset of synthetic but realistic $32 \times 32$ RGB images similar to CIFAR-10, generated using a generative model.

2. A binary classification task on the SVHN dataset (Netzer et al., 2011) with the labels being the parity of the digit, denoted by *SVHN-parity*. For the training data we use a balanced random subset of 'train' and 'extra' partitions, for test data we use the 'test' partition.

We focus on the CIFAR-5m-bin experiments in the main body. Corresponding SVHN-parity experiments can be found in Appendix F.

We use these particular datasets because we need datasets with a large number of samples in order to measure data-scaling, and CIFAR-5m-bin and the SVHN dataset both have $\geq$ 600k samples. We chose to consider binary tasks as this makes the kernel experiments computationally feasible. Although there are other datasets with similar sample sizes (e.g. ImageNet), the datasets we use have the advantage that they

are low-resolution and an easier task— thus, scaling-law experiments are far more computationally feasible. We also do some experiments on a synthetic dataset in Appendix E.

**Architectures.** We use the following base architectures: Myrtle CNN (Page, 2018; Shankar et al., 2020) for the CIFAR-5m-bin task and a 5 layer CNN with 64 channels for the SVHN-parity task. We consider various width scaling for these networks: For the Myrtle CNN we vary the width from 16 to 1024 and from 16 to 4096 for the 5 layer CNN. See Appendix B for more details.

**Experimental Details.** We describe some subtleties in the experimental setup. We use NTK parameterization (Jacot et al., 2018) for both the neural network and the kernels as this is the parameterization used in proving the equivalence of neural network and NTK at infinite width. We train with MSE loss and $\pm 1$ labels. We use test error as the metric for all the plots except in Appendix I where we recreate some of the most important plots for test loss. All of our networks are in the overparameterized regime i.e. are able to reach 0 train error. To preserve the correspondence between the neural networks, empirical NTKs and infinite NTKs we train all of them with SGD with momentum with the same hyperparameters (Appendix B.4). This also ensures that in all experiments neural networks will be trained below the *critical learning rate*, i.e. the learning rate at which training of the empirical and infinite NTK can converge (Appendix B.3). Training for the empirical NTK is done by linearizing the initialized neural neural network using Novak et al. (2020) library while for infinite NTK we directly use SGD with momentum on the linear system given by the infinite NTK and the labels. We describe further experimental details for each individual experiment in Appendix B.

## 3 Data Scaling Laws of Neural Networks and NTKs in the Overparameterized Regime

In this section we compare the data-scaling laws of neural networks to their corresponding emperical NTKs and infinite NTKs. Our main claim is the following.

**Claim 3.1.** *There exists a natural setting of task and network architecture such that the neural network trained with SGD has a better scaling constant than its corresponding infinite and empirical NTK at initialization. Further, this gap in scaling continues to hold over a wide range of widths and learning rates used in practice.*

The above claim can be interpreted as stating that there exists natural settings where the regime in which real neural networks are trained is meaningfully separated from the NTK regime, and real neural networks have a better scaling law.

In Figure 1 (A), we train a Myrtle CNN (Page, 2018; Shankar et al., 2020), its empirical NTK at initialization, and its infinite NTK on the CIFAR-5m-bin task. In each case, we train to fit the train set with SGD and optimal early stopping. We then numerically fit scaling laws, and find, scaling-exponents $\beta$ of: .185 (empirical NTK), .213 (infinite NTK), .291 (neural network). Thus, in this image-classification setting, the real neural network significantly outperforms its corresponding NTKs with respect to data-scaling. We show the statistical significance of this result in Appendix A. See the Appendix B for full experimental details.

We now investigate how robust this result is to changes in the width of the architecture and optimizer, within realistic bounds.

**Effect of Width.** We explore the effect of width. In Figure 2(a) we train neural networks with widths much smaller (16) and much larger (1024) than the width (64) used in Figure 1(A). We find that these networks behaved similarly with respect to their scaling constants (.276 and .279 respectively), and performed better than the infinite width NTK (scaling constant: .213), confirming that *real neural networks are far from the NTK regime.* However, we know that in the infinite width limit, all these methods will perform identically. Moreover, as mentioned in Section 2 we are careful to ensure this limit is preserved by our optimization and initialization setup. This implies that at some point, increasing width of the real network will start to *hurt* performance— although it may be computationally infeasible to observe such large widths. To explore the width-dependency, in Figure 2(b) we plot the expected performance of empirical NTK and Neural network as we *increase the width,* using a fixed training size of 4000. Here we see that (a) the empirical NTK at

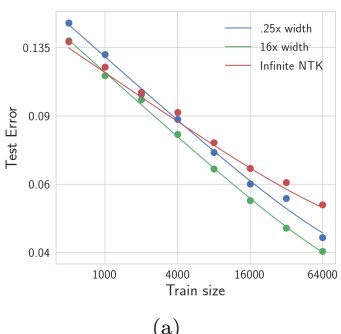 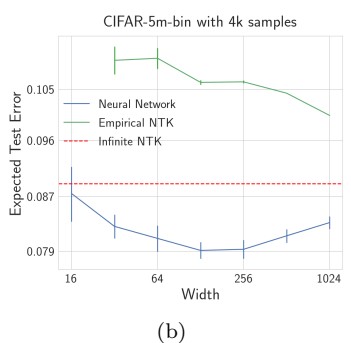 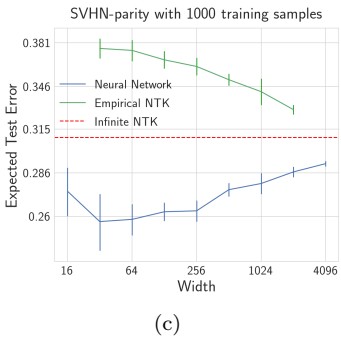

| (a) | (b) | (c) |

Figure 2: On the Effect of width. In Figure 2(a) we plot data scaling laws of the Myrtle-CNN at small (16) and large (1024) widths and the infinite NTK. We observe that both finite widths have similar scaling constant which is better than that of the infinite NTK. In Figure 2(b) we plot the performance of Myrtle-CNN and its empirical NTK for a fixed training size while varying width. In Figure 2(c) we do the same for a 5-layer CNN and the SVHN-parity task. Both Figure 2(b) and 2(c) we observe that (a) the empirical NTK performance continues to improve with width, moving towards the infinite NTK performance while (b) neural network performance improves initially and then starts to deteriorate towards the infinite NTK performance. Error bars represent estimated standard deviation. See Appendix B for more details.

initialization continues to improve with larger width and approaches the infinite NTK's error from above, while (b) the neural network improves initially and then starts to deteriorate and approach the infinite NTK's error from below. In Figure 2(c) we repeat the experiment for the SVHN-parity task. In this setup it was computationally feasible to try out much larger width (upto 4096) with a smaller training size of 1000. Hence in this experiment as the width increases, we can observe stronger deterioration of the performance of neural network, towards the infinite NTK performance.

Together these results suggest that "intermediate" widths (not too large, not too small) are important for the performance of overparameterized neural networks, and any explanatory theory must be consistent with this.

**Effect of Learning Rate.** We now study how robust our results are to changes in the learning rate, within practically used bounds. Note that changing the learning rate only affects the neural network training, and does not affect any of their corresponding NTKs. In Figure 3(a) we train networks in the same setting as Figure 2, but with varying learning rates. We find that after moderate modifications of the learning rate the neural network still has a better scaling law than infinite and empirical NTK at initialization, suggesting that practically used learning rates (for practically used widths) are far from the NTK regime. The scaling constants are .333, .262, .213 for the 3x higher learning rate, 10x lower learning rate and the infinite neural network. We discuss the effects of more drastic changes (1000x) in the learning rate in Appendix C.

**Other Changes in Optimization.** We now study whether our results hold under other changes to optimization parameters. In Figure 3(b), 3(c), 3(d), we see the effect of doing GD instead of SGD, effect of training without momentum and of using the final test error instead of doing optimal early stopping respectively. We see that in all of these cases, while there is some change in the scaling laws, the neural network scaling constant is still always better than the one for infinite NTK. The scaling constants for neural networks in Figure 3(b), 3(c), 3(d) are .294, .310 and .292 respectively. The scaling constant for the infinite NTK is .213 in Figure 3(b), 3(c) and .219 for Figure 3(d). This suggests that these optimization factors (within commonly used values) are not the fundamental reason behind the improved scaling laws of neural networks.

Various extensions to the NTK regime have been proposed (Roberts et al., 2021; Andreassen & Dyer, 2020) in the literature which allow for the change in empirical NTK but posit that higher order analogues of the NTK remain constant. This would predict that higher order analogues of the empirical NTK at initialization would be sufficient to match the performance of neural networks. In Appendix D we show that this is not

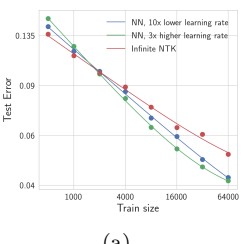 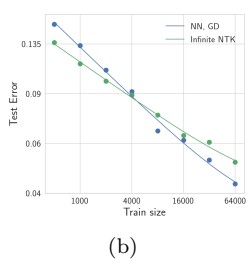 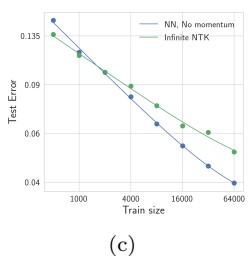 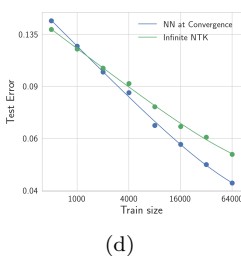

(a)             (b)             (c)             (d)

Figure 3: **Neural networks continue to have a better scaling constant under various hyperparameter choices.** We compare the data-scaling for a Myrtle-CNN, its empirical NTK and its infinite NTK on CIFAR-5m-bin task under various hyperparameter changes: (a) Higher and lower learning rate compared to Figure 1 (b) GD instead of SGD (c) SGD without momentum (d) Training until convergence (no early stopping)

the case suggesting that these theories may also not be sufficient to explain the performance of practical neural networks.

**Discussion and Future Questions.** The equivalence between neural networks and corresponding NTKs applies when width $\gg$ train-size. On the other hand nearly all overparameterized networks and natural tasks fall in the regime of width $\ll$ train-size (though width is still large enough to fit the dataset). The results of this section — showing separations between neural networks in the latter regime and NTKs lead to following concrete question on the gap between theory and practice which could guide future work.

**Question 3.1.** *How can we understand the the behavior of overparameterized networks in the width $\ll$ train-size regime?*

## 4 Exploration of After-Kernel wrt Dataset size

In the previous section, we studied the empirical NTK when linearized around weights at *initialization*. In this section we will study the behaviour of empirical NTK when linearized around the weights obtained at the *end of training*. This is known as the *after-kernel* for empirical NTK, in the terminology of Long (2021). We will show, in the more precise sense defined below, that (1) the after-kernel continues to improves with dataset size, and thus (2) no fixed-time after-kernel is sufficient to capture the data scaling law of its corresponding neural network.

Formally, denote the after-kernel from the neural network trained on $m$ samples as $K_m$. We will denote the accuracy of $K_m$ when fit on $n$ samples as $K_m(n)$. Here, the $n$ samples are a subset of the original $m$ samples. When we use fresh $n$ samples to fit we use the notation $K_m^F(n)$. We study the after-kernel as improved performance of neural networks over NTKs has been attributed (Ortiz-Jiménez et al., 2021; Atanasov et al., 2021) to the adaptation of the empirical NTK of the neural network to the task. Concretely, prior works (Long, 2021; Paccolat et al., 2021) have shown that this explanation is complete in the following sense: The behaviour of $K_n(n)$ is similar to that of the neural network fit on $n$ samples. In other words, when we fit an after-kernel obtained from training on $n$ samples to the same $n$ samples we get an accuracy very close[1] to that of the neural network fit on the same $n$ samples. We verify this for our setup in Figure 4(a). This tells us that the following two factors are sufficient to explain the behaviour of neural networks fit on $n$ samples: (1) Change in empirical NTK from empirical NTK around initial weights to the after-kernel due to training on $n$ samples. (2) Fitting the after-kernel on $n$ samples.

What this does not tell us is how these two improvements scale with training size $n$. In particular, we know that $K_0(n)$ i.e. the empirical NTK at initialization fit on $n$ samples does not match the neural network trained on $n$ samples on the other hand $K_n(n)$ does. This raises the following natural question: *How data*

---

[1]We again note that empirical NTK *does not* refer to the linearization of the network (See Section 2 for an exact definition). If we had linearized the network this statement would be trivially true as the linearized network around final weights would start out with an accuracy matching that of the trained neural network.

*dependent does the kernel need to be to recover the performance of the neural network?* For example, it is possible that for some sample size $m_0$ and all $m \geq m_0$, the after-kernel $K_m$ is roughly constant, and has same scaling law as the neural network itself. We find that this is not the case– the after kernel continuously improves with dataset size $m$.

### 4.1 Experimental Results

**After-Kernel continues to improve with dataset size.** In Figure 4(b) we plot $K_m(500)$ versus $m$ for our base Myrtle CNN from Figure 1. We observe that $K_m(500)$ improves as $m$ goes from 500 to 1024$k$ showing that the after-kernel keeps improving with larger dataset sizes. Corresponding SVHN-parity experiments can be found in Appendix F.

**Fixed after-kernel is not sufficient to capture neural network data scaling.** In Figure 4(b) we plot the data scaling curves for the base Myrtle-CNN, its empirical NTK at initialization, $K_{16k}$, $K_{64k}$ with scaling constants $.291, .185, .103, .097$ respectively. We find that the neural network has the best scaling constant. This shows that the scaling of the after-kernel with training size is an important component of neural network scaling laws as even the after-kernel learnt with 64k samples (on the simple CIFAR-5m-bin task) is not sufficient to explain the data scaling of neural networks. We also see that $K_{64k}$ has better performance than $K_{16k}$, another evidence towards the fact that after-kernel improves with dataset size.

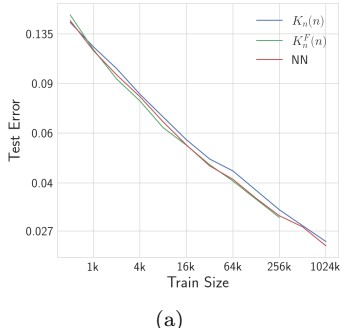
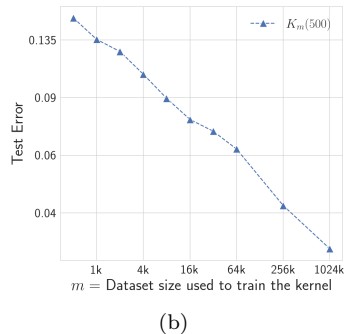
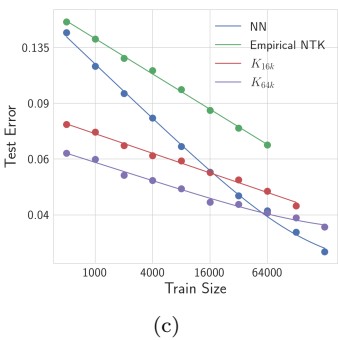

(a)               (b)               (c)

Figure 4: **After-Kernel continues to improve with dataset size.** In Figure 4(a) we plot data scaling curves of $K_n(n), K_n^F(n)$ and the neural network and observe that they behave very similarly. In Figure 4(b) we plot $K_m^F(500)$ versus $m$ and observe that the performance improved with increasing $m$. In Figure 4(c) we plot data scaling curves of empirical NTK at initialization, $K_{16k}$, $K_{64k}$ and the neural network. We observe that the neural network has the best scaling law amongst these.

## 5 Time Dynamics

In the previous section, we saw that the change in the empirical NTK from initialization to the end of training (the *after-kernel*) is sufficient to explain the improved performance of neural networks. Thus the empirical NTK must have evolved throughout training, and in this section we take a closer look at this evolution. Our main focus in this section is to investigate the following informal proposal in the literature (Fort et al., 2020; Long, 2021) about how the empirical NTK evolves:

**Hypothesis 5.1** ((Informal, from Fort et al. (2020); Long (2021))**.** *The empirical NTK evolves rapidly in the beginning of training ($< 5$ epochs), but then undergoes a "phase transition" into a slower regime.*

One way to interpret the above hypothesis is that there is both a qualitative and quantitative difference in the empirical NTK between the "early phase" of training (the first few epochs) and the later stage of training. This is called a "phase transition" in the literature, in analogy to physics, where systems undergo discontinuities between two regimes with quantitatively different dynamics.

In this section we will give evidence that suggests, contrary to prior work, that there is no such "phase transition". We show that if empirical NTK performance is measured at the appropriate scale, performance

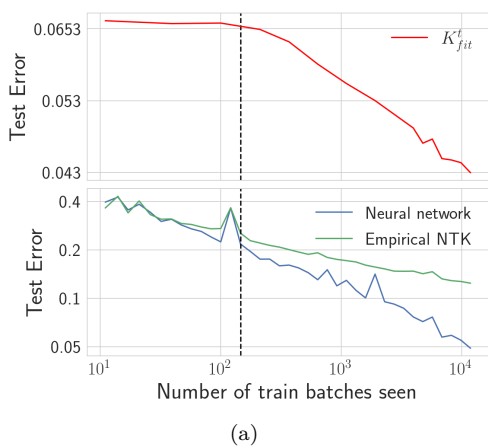 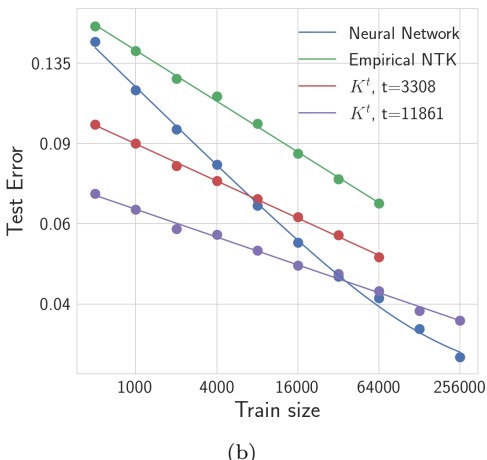

Figure 5: **Empirical NTK keeps improving uniformly throughout most of the training.** In Figure 5(a) we plot the test error of Myrtle-CNN, its empirical NTK at initialization and $K_{fit}^t$ at time $t$. We observe that the slope $K_{fit}^t$ does not decrease with time suggesting that the change in kernel does not slow down after an initial part of training. Using this same setup, we plot the data scaling curves of $K^t$ for various $t$ and the data scaling of Myrtle-CNN in Figure 5(b). We observe that the Myrtle-CNN has the best scaling law.

appears to continuously improve throughout training (from early to late stages), at approximately the same "rate." Our experiments are in fact compatible with the experiments in prior work (e.g. Fort et al. (2020)): we simply observe that if performance and time are measured on a log-log scale (as is appropriate for measuring multi-scale dynamics), then the NTK is seen to improve continuously throughout most of the training.

## 5.1 Experiments

**Setup.** We now describe the setup more formally. Let $K^t$ refer to the empirical NTK (as described in Section 2) extracted at time $t$ in the training, where we measure time in terms of number of SGD batches seen in optimization. $K_{fit}^t$ denotes the model $K^t$ fit to the whole training data. Prior works (Fort et al., 2020; Long, 2021) have used the slope of the curve of test error of $K_{fit}^t$ versus $t$ to decide if the kernel is changing rapidly or not. We will do the same with one crucial difference: We will measure this slope on a log-log plot instead of directly plotting test error and time. We do this as empirically scaling laws with respect to time (or tokens processed) have been observed (Kaplan et al., 2020) for natural language tasks in neural networks and formally proved for kernels (Velikanov & Yarotsky, 2021) for natural tasks. These results suggest the need for log-log plots to observe qualitative phase transitions in training dynamics.

**Results.** Our main claim is that the test error of $K_{fit}^t$ as a function of time $t$ is approximately linear on a log-log scale, throughout the course of training. Recall that $K_{fit}^t$ is the model obtained by extracting the empirical NTK after $t$ batches of training the real neural network.

In Figure 5(a) we compare the test error of the base Myrtle-CNN at time $t$, test error of empirical NTK at initialization at time $t$ and $K_{fit}^t$ when trained on $64k$ samples with the same hyperparameters as Figure 1. Since we want to probe Hypothesis 5.1, which is about the beginning of training, we plot these quantities until train error reaches $< 5\%$ (which requires 32 epochs in our experiments). This should be sufficient to cover any reasonable definition of "beginning of training".

Observe that in Figure 5(a) we do not observe a "phase transition" after which the improvement in kernel test error (in red) slows down. In fact, we observe that the kernel starts out being essentially constant and then starts and continues to improve uniformly.

We instead observe the following two regimes: **(1.)** In the first regime (before the dashed vertical line) the empirical NTK at initialization and the neural network have very similar behaviour, and $K_{fit}^t$ is nearly constant. This only lasts for around 140 batches $\approx 0.5$ epochs[2]. **(2.)** In the next regime (after the dashed line) the empirical NTK at initialization and the neural network diverge. As they diverge, the extracted kernel $K_{fit}^t$ also starts to improve with a constant slope, and this improvement continues uniformly until the terminal stage of training.

Importantly, the kernel $K_{fit}^t$ does not transition into a "slower phase" of learning at any point[3] in our experiments. Corresponding SVHN experiments can be found in Appendix F.

Next, we measure the performance of $K^t$ in terms of its *data scaling law*. Due to computational limitations (since measuring data-scaling is expensive), we can only measure the scaling law for several selected values of time $t$, instead of every batch (as in Figure 5(a)). In Figure 5(b) we plot data-scaling of $K^t$ for $t = 0$ (empirical NTK at initialization), $3308$ (10 epochs), $11861$ (32 epochs), in the same setup as Figure 5(a). We also plot the data scaling of the base Myrtle-CNN with the same hyperparameters. As in Figure 4(c) of Section 4 we again observe that the neural network has the best scaling law, outperforming any of the extracted kernels. This shows that representations learnt after any constant time $t$ of training are not sufficient to explain the data scaling of neural networks. Rather, these representations improve throughout training, and the entire course of training must be considered to recover the correct scaling law.

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

## A   Statistical Significance of Figure 1(A)

In Figure 1(A), for each of the dataset sizes $500, 1000, 2000, \ldots, 64000$ we have $>= 4$ neural network runs. Taking our distribution to be the cross product over there runs and dataset sizes we calculate the distribution of scaling constants of the neural network. We get that the mean is .292 and the standard deviation is .0106. The infinite NTK scaling constant (.213) is 7.5 standard deviations away. The mean and the standard deviations were calculated with a sample size of 10000. Even the minimum scaling constant across these 10000 samples was .265 which is closer to the neural network mean scaling constant (.292) than it is to the infinite NTK scaling constant (.213). The empirical NTK scaling constant: .185 is even smaller and hence the result is even more statistically significant (10.5 standard deviations away).

## B   Experimental Details

### B.1   Architecture

The exact architecture of the Myrtle-CNN we use is the following: 3 Conv Layer with $c$ channels, Relu, 3 Conv Layer with $2c$ channels, Relu, $2 \times 2$ Avg-pooling, 3 Conv Layer with $4c$ channels, Relu, $2 \times 2$ Avg-pooling, 3 Conv Layer with $8c$ channels, Relu, $2 \times 2$ Avg-pooling, $4 \times 4$ Avg-pooling, Dense Layer with 1 output. Stride is always 1. Here is our code for the network in the *stax* module of the Neural Tangents (Novak et al., 2020) library:

```
init_fn, apply_fn, kernel_fn = stax.serial(
    stax.Conv(c, (3, 3), strides=(1, 1), padding='SAME'), stax.Relu(),
    stax.Conv(2*c, (3, 3), strides=(1, 1), padding='SAME'), stax.Relu(),
    stax.AvgPool((2,2), (2,2)),
    stax.Conv(4*c, (3, 3), strides=(1, 1), padding='SAME'), stax.Relu(),
    stax.AvgPool((2,2), (2,2)),
    stax.Conv(8*c, (3, 3), strides=(1, 1), padding='SAME'), stax.Relu(),
    stax.AvgPool((2,2), (2,2)),
    stax.AvgPool((4,4), (4,4)),
    stax.Flatten(),
    stax.Dense(1)
)
```

We refer to $c$ as the "width" of the network. Our base network in Figure 1 has $c = 64$.

We use the following 5 layer CNN for the SVHN-parity task:

```
init_fn, apply_fn, kernel_fn = stax.serial(
    stax.Conv(c, (3, 3), strides=(1, 1), padding='SAME'), stax.Relu(),
    stax.Conv(c, (3, 3), strides=(1, 1), padding='SAME'), stax.Relu(),
    stax.Conv(c, (3, 3), strides=(1, 1), padding='SAME'), stax.Relu(),
    stax.Conv(c, (3, 3), strides=(1, 1), padding='SAME'), stax.Relu(),
    stax.Flatten(),
    stax.Dense(1)
)
```

The base network has $c = 64$.

The MLP that we use in our synthetic dataset experiments is a depth-4 MLP with the following code:

```
init_fn, apply_fn, kernel_fn = stax.serial(
    stax.Flatten(),
    stax.Dense(c), stax.Relu(),
    stax.Dense(c), stax.Relu(),
    stax.Dense(c), stax.Relu(),
```

```
    stax.Dense(1)
)
```

## B.2    Scaling Laws

In our plots we use the scaling laws for the form $L(n) = A(1/n + \alpha)^\beta$ where $\beta$ is referred to as the scaling constant. This is take into account the fact that any given neural network has a maximum possible accuracy. Note that as our task is a deterministic there is no label-noise to be accounted for. We calculate $A, \alpha, \beta$ by solving the least squares problem between $\log(L(n))$ and the log of empirically found test errors.

## B.3    SGD with momentum, equivalence between infinite NTKs and neural networks

The standard equivalence between neural networks and infinite NTKs (Lee et al., 2019) used GD with no momentum. Recent works (Dyer & Gur-Ari, 2020; Liu & Pan, 2021) have extended this to SGD and SGD with momentum.

## B.4    Kernel-SGD

We optimize our neural network with SGD, the empirical NTK (as the the linearized neural network) with SGD and the infinite NTK with kernel-SGD (described later), all with the same optimization hyperparameters. For all of the experiments we use early stopping except in Figure 3(d). This allows us to ensure that the limiting behaviour of all 3 models as width tends to infinity will exactly be the same in contrast with prior works which have usually optimized these 3 models in different ways. This is important for having a fair comparison between all three models.

We now describe kernel-SGD. Intuitively, kernel-SGD refers to doing SGD in the kernel space. We now define it more formally, beginning with the notation. Let

- $n$ denote the number of samples.

- $K = FF^T$ where $K$ is the $n \times n$ NTK matrix and $F$ is a $n \times d$ feature matrix. The row of $F$ corresponding to training sample $x$ is given by $f(x)^T$ where $f$ is the function that maps a training sample to their features. Define $k(x)$ as $Ff(x)$ and $k(x, x')$ as $f(x)^T f(x')$

- $y_x$ be the label corresponding to sample $x$.

- $\|Fw - y\|^2$ be the loss function. As we will be doing SGD start with $\mathbf{0}$ initialization we can equivalently state the loss function as $\|Kz - y\|^2$ where $w = F^T z$.

Kernel-SGD refers to doing SGD on the loss $\|Kz - y\|^2$, optimizing for $z$. Specifically if our batch contains a single sample $x$ then we do one step of GD on the loss $\|k(x)^T z - y_x\|^2$. See Blondel (2012) for an implementation.

In the following claim we show that this is equivalent to doing SGD on the loss $\|Fw - y\|^2$ as long as our selection of the the batches and SGD hyperparameters is the same. The following claim only works for one step and SGD with batch size 1 but can be easily generalized. The result is easy to prove and probably follows from prior works. But we were not able to find a reference so we present it here for completeness.

**Claim B.1.** *The prediction on a input sample $x'$ is the same in the following two cases*

- *We start from $w = F^T z_0$ and do one step of SGD on input sample $x$ in the feature space i.e. one step of GD with loss $(f(x)^T w - y_x)^2$. The learning rate is $\eta$.*

- *We start from $z = z_0$ and do one step of SGD on input sample $x$ in the kernel space i.e. one step of GD with loss $(k(x)^T z - y_x)^2$. The learning rate is $\eta$.*

*Proof.* The update in the first case leads to

$$w' = w_0 - 2\eta(f(x)^T w_0 - y_x)f(x) = F^T z_0 - 2\eta(f(x)^T F^T z_0 - y_x)f(x) = F^T z_0 - 2\eta(k(x)^T z_0 - y_x)f(x)$$

The prediction on sample $x'$ changes to

$$f(x')^T w' = f(x')^T(F^T z_0 - 2\eta(k(x)^T z_0 - y_x)f(x)) = k(x')^T z_0 - 2\eta k(x, x')(k(x)^T z_0 - y_x))$$

The update in the second case leads to

$$z' = z_0 - 2\eta(k(x)^T z - y_x)k(x)$$

The prediction on sample $x'$ changes to

$$k(x')^T z' = k(x')^T(z_0 - 2\eta(k(x)^T z - y_x)k(x)) = k(x')^T z_0 - 2\eta k(x, x')(k(x)^T z_0 - y_x))$$

which is exactly the same as the update in the first case.

$\square$

## B.5 Additional Experimental Details for Figures

**64 bit versus 32 bit precision** Our neural network experiments are done with 32 bit precision while kernel experiments are done with 64 bit precision. To verify that this difference is not the cause of better performance of neural networks we ran the experiment in Figure 1 with 64 bit precision for train size of 64k. The test error actually improved, test error with optimal early stopping was .03993 while it was .04118 at 32 bit precision.

**Starting with 0 output neural networks** The convergence between neural networks, empirical NTK and the infinite NTK as width tends to infinity needs that the neural networks have 0 output at initialization. This can be done by subtracting the initial outputs from the neural network output. While we do not do this for the experiments in the paper, we verify in Table 1 that this makes almost not difference for Figure 1. We use a table instead of a plot as most of the differences are too small to be visible on a plot.

| Dataset Size | 500 | 1k | 2k | 4k | 8k | 16k | 32k | 64k |
|---|---|---|---|---|---|---|---|---|
| Usual Neural Networks | .1504 | .1177 | .09655 | .08083 | .0658 | .0545 | .04958 | .04118 |
| Neural Netwoks with 0 output | .1501 | .1204 | .09672 | .08302 | .0668 | .05462 | .04585 | .04152 |

Table 1: Performance of usual neural networks and their 0 output counterparts.

We now describe additional experimental details for each figure beyond those described in Section 2.

In Figure 1 all 3 models were trained with batch size of 200 and SGD with learning rate of 4.0 and momentum of .9 (as implemented in *jax.experimental.optimizers.momentum*). For all models we used optimal early stopping where the test error is logged after an increase of 1.1 multiplicative factor in the number of gradient steps. Unless mentioned otherwise this will the optimization setup we will use. For neural network and empirical NTK each model is averaged over 4 random initializations and error bars denote standard deviations (except the last two points of empirical NTK which we were not able to rerun due to computational constraints). The infinite NTK is a deterministic model hence we only have only have a single run for it. We also trained the kernels by directly solving the linear system with optimal L2-regularization but that yielded worse test performance.

In Figure 2(b), for empirical NTK with large widths (128, 256) the values across different initializations were very similar hence we did not do multiple runs of even higher widths (512, 1024). For the neural network standard error of the mean (SEM) was calculated using 6 runs for widths up to 128 and 4 runs for higher widths.

In Figure 2(c) the models are trained with SGD: batch size of 200, learning rate of 10, .9 momentum. All models are averaged across 8 random initializations. In this figure we report the final test error at convergence (train loss $\leq 2 \times 10^{-6}$ for neural network) for all models.

In Figure 3(c) we train with learning rate of 10.0 with no momentum.

In Figure 3(d) we train the width 512 version of the Myrtle-CNN till it reached train loss $\leq .001$ by which point their test error converges to a nearly fixed value.

In Figure 4 all after-kernels are extracted at the optimal early stopping point of training. In Figure 4(a) and 4(b) the train sizes used are $500, 1k, 2k, 4k, 8k, 16k, 32k, 64k, 256k, 1024k$.

In Figure 5 with fit $K^t$ to the data with same optimization hyperparameters used in Figure 1: batch size 200, SGD with learning rate 4.0 and momentum .9.

## C Effect of Very Low Learning rate

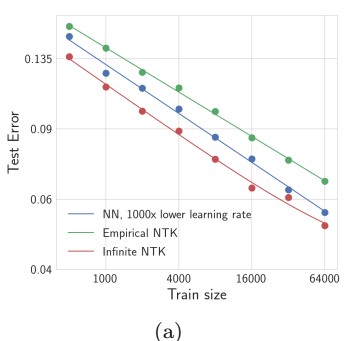 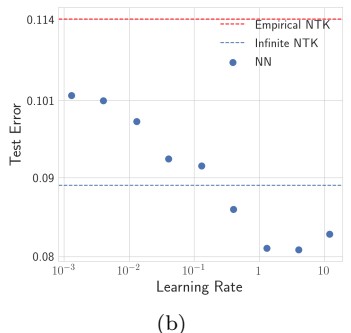 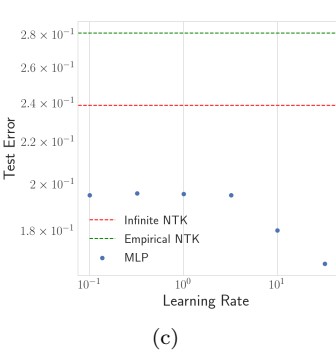

|  |  |  |
|---|---|---|
| (a) | (b) | (c) |

Figure 6: In Figure 6(a) we redo the experiment from Figure 1 except that training the neural network with a 1000x smaller learning rate. This leads to a significant drop in the performance and the scaling constant of the neural network. In Figures 6(b) and 6(c) we explore the effect of learning rate on a fixed training size.

In this section we explore the effects of very low learning rate. The motivation is to understand the gradient flow limit i.e. the limiting behaviour of trained neural networks as the learning rate tends to 0. In Figure 6(a) we repeat Figure 1 except that we train the neural network with learning rate .004. The measured scaling constants are reasonably close by: $.204, .185, .213$ for the Myrtle-CNN, its empirical NTK and infinite NTK respectively. This suggests a natural question:

**Question C.1.** *Do the benefits (with respect to scaling constant) of finite width networks over corresponding empirical NTK vanish in the gradient flow limit?*

To answer this question affirmatively we would would need to repeat the plot of Figure 6(a) for various widths which was computationally infeasible for us. We leave this for future work.

We now move to considering the effect of learning rate on a fixed training size.

In Figure 6(b) we plot the performance with respect to learning rate for training sizes 4000 (From the setup in Figure 1) and observe that at low learning rates performance is worse[4] than infinite NTK but still better than empirical NTK at initialization. From this plots it is not clear if at the lowest learning rates which we could train the performance has converged to the gradient flow performance. In this setup it was computationally infeasible for us to explore smaller learning rates. To do this we move to the synthetic setting (with the same setup as in Figure 8) in Figure 6(c). Here the performance (final test error) converges as we go towards smaller learning rates which indicates that we have converged to the gradient flow limit. In this limit the neural network performs better than the infinite and empirical NTK at initialization. Note that higher learning rates still leads to even better performance.

We interpreted all of these experiments as suggesting that **while high learning rate plays an important role in the performance of empirical networks it may not be necessary in having a improved**

---
[4]Note that this is just for a single width. We do not know how width affects these results. We do know that at infinite width the learning rate does not have any effect, other than due to optimal early stopping.

**performance over corresponding NTKs.**. But more experimental evidence is needed to understand the role of learning rate and in understanding the gradient flow limit, particularly for natural tasks.

**Related Works:** Barrett & Dherin (2021) and Smith et al. (2021) describe regularizers which potentially explain the benefits of high learning rate in GD and SGD. The effect of higher than critical learning rate on very wide networks has been studied by Lewkowycz et al. (2020). They claim that perhaps the improved generalization of neural networks can be explained by the use of higher than critical learning rate. This does not apply to our experiments as all of our experiments use learning rates which are below the critical learning rate.

## D   Higher order analogues of the NTK

Let $f(w, x)$ with $w$ representing the weights and $x$ a sample. By Taylor expansion around $w_0$ we have:

$$f(w, x) = f(w_0, x) + \nabla_w f(w, x)|_{w_0}(w - w_0) + \frac{1}{2}(w - w_0)^T \nabla_w^2 f(w, x)|_{w_0}(w - w_0) + \dots$$

The empirical NTK of the neural network around weights $w_0$ refers to the model $g_1(w, x) = \nabla_w f(w, x)|_{w_0}(w - w_0)$.

We consider the following two second order analogue of the NTK: $g_2^{only}(w, x) = \frac{1}{2}(w - w_0)^T \nabla_w^2 f(w, x)|_{w_0}(w - w_0)$ and $g_2^{full}(w, x) = f(w_0, x) + \nabla_w f(w, x)|_{w_0}(w - w_0) + \frac{1}{2}(w - w_0)^T \nabla_w^2 f(w, x)|_{w_0}(w - w_0)$

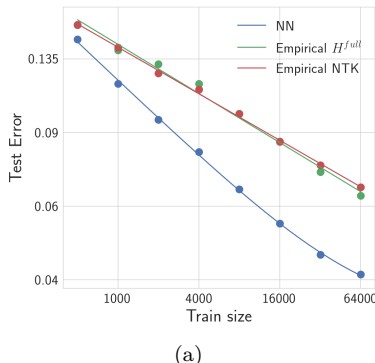

(a)

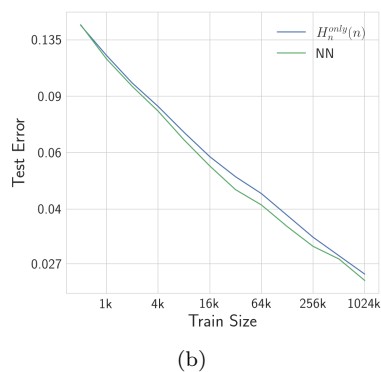

(b)

Figure 7: Higher order analogues of Empirical NTK.

In Figure 7(a) we use the setup of Figure 1 to plot the performance of the $2^{nd}$ order analogue of the empirical NTK ($g_2^{full}$, referred by $H^{full}$ in the plot). This shows that even this higher order analogue is not sufficient to recover the scaling law of neural networks.

In Figure 4(a) we saw that the after-kernel was sufficient to explain the improved performance of neural network over the empirical NTK. We now show an analogous result for the $2^{nd}$ order analogue of NTK. As we want to understand the effect of change in the higher order terms we need to remove the influence on the after-kernel. We do so by defining the higher order analogue of after-kernel as the model $g_2^{only}$ which does not contain the after-kernel. We will denote this by $H_m^{only}$ when we use the weights after training on $m$ samples. In Figure 7(b) we show that the performance of $H_n^{only}(n)$ is very close to that of the neural network.

Both of these experiments suggest that theories which assume that higher order analogues of the NTK remain fixed throughout the training may not be sufficient to explain the performance of neural networks.

# E    Experiments on Synthetic Data

Some of our experiments were not feasible on the CIFAR-5m-bin and SVHN-parity tasks. We did these experiments on the following synthetic task: Sample $z \sim \{-1, 1\}^{30}$, $\epsilon \sim \mathcal{N}(0, .25 \cdot I_{30})$, the input sample is $x = z + \epsilon$ and the label is $y = z_1 z_2$.

Experiments on very low learning rates for this synthetic task can be found in Appendix C.

In Figure 8 we do an analogous experiment to Figure 2(b) and 2(c) for this synthetic task. We again observe that neural networks at small width improve with increase in width but at high width they start to worsen off with increase in width. The models are trained with SGD: batch size of 100, learning rate of 10, no momentum. All models are averaged across 8 random initializations. In this figure we report the final test error at convergence (train loss $\leq 2 \times 10^{-6}$ for neural network) for all models.

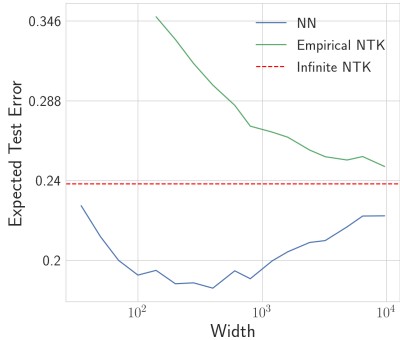

Figure 8: Analogous experiment to Figure 2(b) and 2(c) for the synthetic task

# F    SVHN-parity Experiments

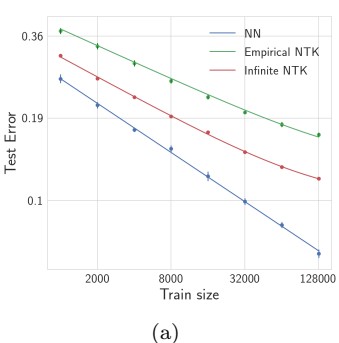
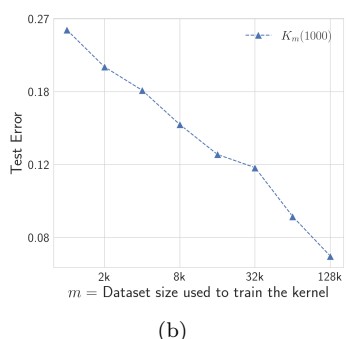
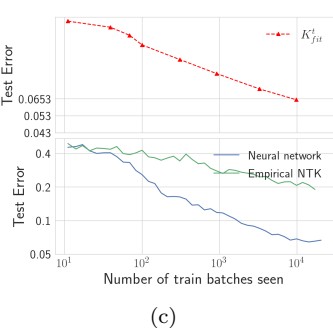

(a)                                    (b)                                    (c)

Figure 9: Experiments for the SVHN-parity task, analogous to Figure 1. Error bars in Figure 9(a) denote standard deviation.

In Figure 9 the analogue of Figure 1. The scaling constants in Figure 9(a) are .28, .22 and .19 for the neural network, infinite NTK and the empirical NTK at initialization respectively. We also observe that unlike the CIFAR-5m-bin task here the neural network outperforms the infinite NTK even at small dataset sizes. This may be because the inductive bias of the NTKs is not suited for a the parity task. In Figure 9(c) we plot the more extensive figure corresponding to Figure 5(a). We again observe that the kernel continues to improve for most of the training.

All of the above models are trained with SGD: batch size of 200, learning rate of 10, .9 momentum. In Figure 9(a) the neural network and empirical NTK experiments are averaged over 4 runs with error bars denoting standard deviation. The infinite NTK is a deterministic model and hence we only do a single run.

Figure 2(c) is another SVHN-parity experiment.

## G  Other related work

**Beyond Double Descent:**  Double descent (Belkin et al., 2019; Geiger et al., 2019; Nakkiran et al., 2020) predicts that in the regime of overparameterized models increasing the width improves the test error. We observe that the performance of overparameterized models is better than that of infinite width models showing that there is a natural setting where there is at least one more ascent after the double descent phenomena. Behaviours beyond the double descent phenomena have been predicted (Adlam & Pennington, 2020; d'Ascoli et al., 2020; Li et al., 2021) and also observed (Lee et al., 2020) in empirical neural networks. Our works is different from these works as we show that in our setup simultaneously a) empirical NTK displays a monotonic improvement in the overparameterized regime towards the infinite NTK performance while b) the neural network performs better than the infinite NTK. This directly points towards another ascent after the double descent and also pinpoints its cause as the divergence between finite width neural networks and the empirical NTK at initialization.

**After Kernel** The empirical NTK after the training of neural network has been termed as *after-kernel* (Long, 2021). It has been shown (Long, 2021; Paccolat et al., 2021). We extend these works by studying how the after-kernel changes with dataset size and show that it continues to improve with dataset size.

**Time dynamics of training from the NTK perspective.** has been studied by Fort et al. (2020); Ortiz-Jiménez et al. (2021); Atanasov et al. (2021); Long (2021). These papers suggest that the empirical NTK changes rapidly in the beginning of the training followed by a slowing of this change. We argue against this interpretation in Section 5.

**Explanations for Scaling Laws** Current explanations of scaling laws (Sharma & Kaplan, 2020; Bahri et al., 2021) rely on a fixed representation space. Operationalizing representation as the after-kernel, our results suggest that in practical neural networks the representation itself improves as data-size increases. Hence we may need more refined theories for explaining scaling laws of neural networks which take this into account.

**Provable differences between neural networks and NTKs** have been shown (Ghorbani et al., 2020; Daniely & Malach, 2020; Karp et al., 2021) though they have been restricted to synthetic datasets.

**Effect of learning rate:** See the last paragraph of Appendix C.

## H  Scaling curves of various after-kernels

In Figure 4(c) we saw that the neural network has a better scaling constant than any fixed after-kernel. Another curious observation from the plot is the comparison between the empirical NTK and $K_{16k}$ scaling curve. We observe that

- $K_{16k}$ outperforms the empirical NTK of all training sizes we tested.

- The empirical NTK has a better scaling constant than $K_{16k}$.

We start by noting that this issue of having worse scaling but always generalizing better does not arise for other comparisons in the paper which involve the neural network. This is because the neural network has better a scaling constant and it also starts to perform better for larger dataset sizes.

Going back to empirical NTK and $K_{16k}$ we can ask the following question: Does empirical NTK having a better scaling constant imply that for large enough sample sizes empirical NTK would outperform $K_{16k}$? Not necessarily, see Figure 10(a) for an example of two scaling curves[5] where the curve with the worse scaling constant (.25) always outperforms the curve with the better scaling constant (.5). We think this will also

---

[5]The irreducible component of the error is necessary for this situation to arise. Irreducible error refers to error that the model will have at $n = \infty$.

hold for empirical NTK and $K_{16k}$. The intuition behind this is that both empirical NTK and $K_{16k}$ have the same number of parameters and both are linear models but $K_{16k}$ has a better implicit bias (as it has better performance[6]).

On the other hand, our current scaling curves do not predict this i.e. extrapolating them predicts that empirical NTK would outperform $K_{16k}$ at $n = 5$ million, see Figure 10(b). But we think this is not a credible prediction for multiple reasons:

- At $n = 5$ million we will have more samples than parameters ($\sim 1.5$ million). This means that we might no longer be overparameterized and hence extrapolating data scaling curves is not valid.

- This could occur due to small fluctuations in the fitting of the scaling curves (particularly as we are extrapolating a lot, factor of 100). Specifically we think that our estimation of irreducible loss for these two curves is noisy.

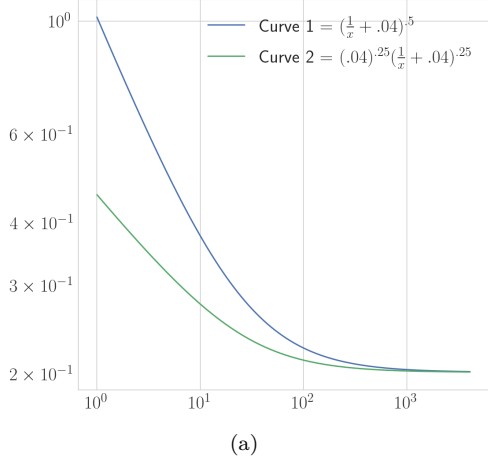

(a)

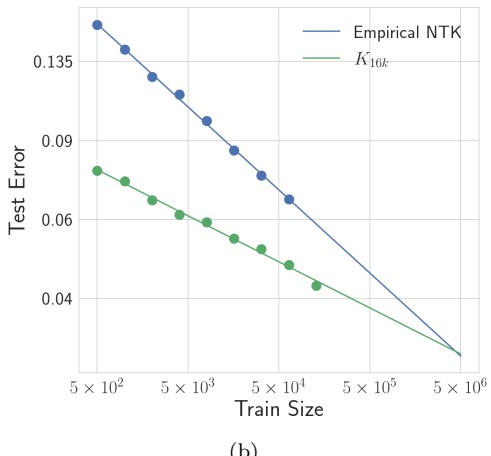

(b)

Figure 10:

# I  Plots with Test Loss

Throughout the paper we have focused on the test error. Here we recreate some of the important plots for test loss instead of test error. These correspond to Figure 1(a), 2(b) and 5(a) respectively. All the phenomena that we observed for test error in these plots continue to hold for test loss.

The scaling constants in Figure 11(a) are .22, .171 and .135 for the neural network, infinite NTK and the empirical NTK respectively.

---

[6]This intuition does not apply to a NTK model and a neural network as they are models of different types, linear and nonlinear respectively.

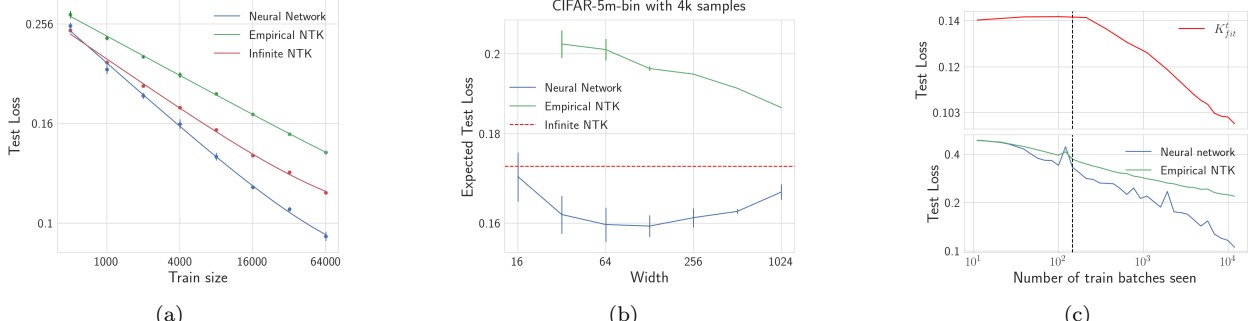

(a)    (b)    (c)

Figure 11: Plotting some of our important figures with test loss as the metric.

