# OpenReview forum: "Empirical Limitations of the NTK for Understanding Scaling Laws in Deep Learning"
_TMLR — Accepted by TMLR_

### Review · Reviewer_Yjp1 · 2023-04-17

**Summary Of Contributions:**

This work empirically shows that the scaling of the generalization error of a finite width network is different from NTK

**Audience:**

Yes

**Claims And Evidence:**

Yes

**Requested Changes:**

1. There are a lot of typos in the references. For example, the first sentence of the paper has "Jacot et al. Jacot et al (2018),"  and similar mistakes appear throughout the paper. The authors should be careful and patient, checking obvious errors thoroughly before submission

2. A lot of figures are not quite visible. For example, Figure 3. The authors should make the font sizes of the figures match the font sizes of the main text. In this regard, all figure (not just Figure 3) can be improved

3. The paper is all about scaling exponents. Why not show the exponents in the figure or in the captions?

4. This is the first major problem: none of the reported scaling exponents have an error bar. While the paper is about saying that the exponents are different, there is no scientific evidence that there is a difference. The authors need to state the error bar and p-value to convince the readers that these measured quantities are significantly different

5. Also, I think the authors need to significantly tone down their claims. Even if the authors show that the exponents are statistically different, the difference is still quite minor in my opinion. For example, for Figure 1, the NTK has 0.213 while the actual NN has 0.291. Supposing both models start from the same performance, how many times more data points are required for the NN to have half of the test error of the NTK? Solving the condition $0.5 * n^{-0.213} = n^{-0.291}$ shows that $n\approx 7000$. We need to have a 7000 times larger dataset to see a significant difference between NTK and actual NN. To be honest, I think this does not convince me (or any ordinary reader) that NN is very different from the NTK. To me, it feels like the empirical evidence only shows that NTK is quite good -- I am not saying that NN is the same as NTK, but that the authors should be much more humble in many claims. For example, in the abstract, the authors call this a "fundamental difference," but I am not convinced. A difference between 0.2 and 2 feels fundamental, but a difference between 0.21 and 0.29 does not.

6. A lot of claims do not feel right. The paper mentioned quite a few times "neural networks provably converge to the NTK at infinite width."
Is there a reference to this? I think a lot of qualifiers are required here, for example, "under gradient flow and special learning rate scaling" and "width certain type of initializations."

7. The term "SGD" in claim 3.1 bepuzzles me. Are the experiments not using GD? Which part of the experiment is using SGD? If SGD is used, a crucial question arises. Is the difference due to finite step size or minibatch training? There is one more parameter to control, and I think the authors should investigate the causes of the difference in greater depth.

**Strengths And Weaknesses:**

The strength of the paper is the systematic empirical exploration of the scaling laws

The weakness is the lack of testing of the statistical significance of the results, and the over-claims associated with it. See the requested changes

---

> ### Author Response · Authors · 2023-05-03
> **Response to Reviewer Yjp1**
>
> >There are a lot of typos in the references. For example, the first sentence of the paper has "Jacot et al. Jacot et al (2018)," and similar mistakes appear throughout the paper...
> >A lot of figures are not quite visible. For example, Figure 3. The authors should make the font sizes of the figures match the font sizes of the main text. In this regard, all figure (not just Figure 3) can be improved
> >The paper is all about scaling exponents. Why not show the exponents in the figure or in the captions?
>
> Thank you for pointing these out, we will do these edits.
>
>
> >This is the first major problem: none of the reported scaling exponents have an error bar. While the paper is about saying that the exponents are different, there is no scientific evidence that there is a difference. The authors need to state the error bar and p-value to convince the readers that these measured quantities are significantly different
>
> We analyze the statistical significance of scaling exponents in Appendix A. We find that the mean scaling constant for the neural network curve is .292 and the standard deviation is .0106. The infinite NTK scaling constant (.213) is 7.5 standard deviations away.
>
> >Also, I think the authors need to significantly tone down their claims. Even if the authors show that the exponents are statistically different, the difference is still quite minor in my opinion. For example, for Figure 1, the NTK has 0.213 while the actual NN has 0.291.
>
> We think that the difference in .291 and .213 is quite significant (see also the next point). The reviewer mentions that a difference of .2 and 2 would feel significant. This is unlikely to be achieved since scaling constants are usually bounded above by 1 and are not above .5 for vision datasets (See Figure 1 top-right in https://arxiv.org/abs/2102.06701).
>
> >We need to have a 7000 times larger dataset to see a significant difference between NTK and actual NN. To be honest, I think this does not convince me (or any ordinary reader) that NN is very different from the NTK.
>
> We don’t think this is the right way to view this since the constant of half is arbitrary. In our opinion the right calculation would be to see how much more data would NTK need to get to the same loss value. Suppose we train the neural network with n=50k (matching CIFAR-10 but much smaller than large scale datasets used nowadays). Then (doing a similar back of the envelope analysis as the reviewer) we would need a  (50k)^(.291/.213-1) > 50 times larger dataset! For larger datasets such as Imagenet we would need a (128*10^4)^(.291/.213-1) > 170 times larger dataset.
>
>
>
> > A lot of claims do not feel right. The paper mentioned quite a few times "neural networks provably converge to the NTK at infinite width." Is there a reference to this? I think a lot of qualifiers are required here, for example, "under gradient flow and special learning rate scaling" and "width certain type of initializations."
>
> As we describe in the beginning of Section 1.1:
> “Our main focus is to understand feature learning occurring due to finite width. To do this, we make the following deliberate choices in all of our experiments: a) We use the NTK parameterization, this makes sure that infinite width networks will be equivalent to kernels b) We use the same optimization setup for 2 Under review as submission to TMLR the neural network, empirical NTK and Infinite NTK, this makes sure that as width tends to infinity all 3 models will have the same limit. We make sure that our comparisons are robust by c) using scaling laws to compare these models and d) doing various hyperparameter ablations (Figure 3).”
>
>
> wrt "under gradient flow and special learning rate scaling" this is not needed for convergence to NTK at infinite width. See theorem 2.1 in https://arxiv.org/abs/1902.06720 where all we need is to use a learning rate below eta_critical which is just the maximum stable learning rate for the infinite width NTK. As described above we make sure to train all networks with the same learning rate (Even the infinite NTK for which we use kernel-SGD as described in B.4) and they all train stably thereby ensuring that our learning rate is below eta_critical.
>
>
> >The term "SGD" in claim 3.1 bepuzzles me. Are the experiments not using GD? Which part of the experiment is using SGD? If SGD is used, a crucial question arises. Is the difference due to finite step size or minibatch training? There is one more parameter to control, and I think the authors should investigate the causes of the difference in greater depth.
>
>
> 1. https://arxiv.org/abs/1909.11304 (Contribution 2) and Section 4 of https://arxiv.org/abs/1909.12292 show that even with SGD neural networks (in NTK initialization) converge to inf width NTK.
> 2. Figure 3b uses GD which we did to ablate wrt SGD and GD. We find that even with GD the neural network has a better scaling constant.
>
> All other experiments use SGD.

---

> > ### Comment · Reviewer_Yjp1 · 2023-05-09
> > **reply**
> >
> > Thanks for the reply. Perhaps the main problem with the work is its very broad title: "Limitations of the NTK for Understanding Generalization in Deep Learning." This work focuses on scaling law and also on the empirical aspect of it. If the paper has such a specific contribution, it should not use a broad title. The authors should emphasize "scaling law" and "empirical" in the title. Something like "Empirical Limitation of the NTK for Understanding Scaling Law in Deep Learning" feels much more appropriate.
> >
> > Also, why is the word "limitations" plural? I only see one specific limitation studied by this work.
> >
> > I am not against accepting this work if the authors change the title to a more specific and meaningful one. The authors do not have to use the one I suggested, but it should match the specificness of my suggestion.

---

> > > ### Author Response · Authors · 2023-05-09
> > > **reply**
> > >
> > > We think of a scaling laws as a tool to understand generalization behavior of neural networks. As we discuss in the paper we think this is a stronger test than a comparison of generalization performance for a fixed dataset size.
> > >
> > > >Also, why is the word "limitations" plural? I only see one specific limitation studied by this work.
> > >
> > > Sections 4 and 5 give other limitations of the NTK, namely that even if we use the NTK after training for some time/data it is insufficient to explain the generalization behavior of neural networks.
> > >
> > > We are happy to change the title to "Empirical Limitations of the NTK for Understanding Generalization in Deep Learning.".

---

> > > > ### Comment · Reviewer_Yjp1 · 2023-05-09
> > > > **comment**
> > > >
> > > > There are many aspects of generalization, and scaling law is just one of many potential aspects. For example, the constant term in front of the polynomial term, for many people and for many practical problems, it is the constant term that actually matters. Does the current manuscript show that NTK also predicts this constant wrong? No.
> > > >
> > > > Another aspect is the other end of the spectrum. What happens for a small dataset size? This regime is the non-scaling regime. Does this paper answer anything about this regime? No.
> > > >
> > > > Are there aspects we are not yet aware of? Probably yes.
> > > >
> > > > Which aspects of generalization are the most important? It is difficult to tell, and it is up to the audience. This is why titles of scientific works should faithfully represent their contribution. Now, suppose I write a paper showing that NTK predicts the constant term of generalization wrong. Can I also title my paper "Empirical Limitations of the NTK for Understanding Generalization in Deep Learning"? According to the authors' logic, I can because it is also a tool to probe generalization behavior. However, the problem is obvious: every paper that studies some "tool" to measure the generalization of NTK will be able to title itself "Empirical Limitations of the NTK for Understanding Generalization in Deep Learning" -- this can lead to great confusion in the field.
> > > >
> > > > I insist that the authors be specific about titling.

---

> > > > > ### Author Response · Authors · 2023-05-09
> > > > > **reply**
> > > > >
> > > > > > Which aspects of generalization are the most important? It is difficult to tell, and it is up to the audience.
> > > > >
> > > > > We agree with this but we don't think this implies that "generalization" cannot be used in the title. If we go with this reasoning "generalization" cannot be used in any paper title since a single paper cannot study all aspect of generalization. The abstract of the paper also makes it clear that we are using the lens of scaling laws.
> > > > >
> > > > > Still if the reviewers and editors think this is an important point we can change our title to "Empirical Limitations of the NTK for Understanding Scaling Laws in Deep Learning".

---

### Review · Reviewer_MrBR · 2023-04-22

**Summary Of Contributions:**

The authors study the generalization errors between predictors (i) neural network (ii) after-kernel and (iii) empirical kernel (linear-classifier) in terms of the-called scaling law that is the gen. error as a function of training dataset size. These three predictors converge to the same predictor in the infinite width limit. In the finite width case, the parameters move from initialization and across datasets and architectures neural networks perform better than the after kernel which is better than the empirical kernel. Moreover, unlike the recent results on the stability of the NTK after a few epochs of training, the paper shows that the empirical kernel keeps constant improvement at a linear rate in the log-log scale (beyond the initial phase of training; over 32 epochs).

**Audience:**

Yes

**Claims And Evidence:**

Yes

**Requested Changes:**

The paper is a valuable contribution as it contributes to the study of neural networks through the lens of NTKs. However, I think the paper is too focused on the standard comparison between NTKs and neural networks. It is expected that there is separation for finite-width networks and that after-kernel does better than the empirical kernel. The more interesting insight would be what the scaling laws of NTKs tell us about the performance of neural networks. Are they helpful to predict or understand the generalization of finite-width neural networks?

Follow-up question: Is there any link between the performance of the after-kernel and the evolution speed of ENTK?

Typos:
* emperical NTK (page 1, italic) -> empirical
* Question 3.1: the the (double the)
* truly infinite width -> infinite width (there is only one limit in this scaling)


**Strengths And Weaknesses:**

*Strengths:*

* Scaling exponents are a solid and thorough way of comparing two classifiers in the light of recent studies.
* The findings on the stable rate of evolution of the ENTK are interesting.
* It is interesting that in the small-dataset regime in Figure 2-(b-c) the neural networks of small width perform the best and then deteriorate gradually as the width increases (following the classical U-shape curve and overfitting in the malign sense).

*Weaknesses:*

* It is expected that as feature extractors, neural networks will perform better than a random features linear predictor (NTK at init). It is fair to compare the after-kernel with finite-width NN as done in the previous work. The thing that I find unsatisfactory is why should one use the after-kernel as a linear predictor with learned features, say instead of re-training the last layer of the neural network. I think the more interesting part of the paper is the evolution of the ENTK.
* Overall I would appreciate more interpretations in terms of feature learning and also in general.
* I might be missing something but in Figure 5-a, the caption writes empirical NTK **at initialization**. How is it possible that it evolves during training if it is fixed at initialization? I am guessing that it improves because its training dataset also has grown. Is that correct?
The text needs update accordingly, it is confusing as it is.

---

> ### Author Response · Authors · 2023-05-03
> **Response to Reviewer MrBR**
>
> >It is expected that as feature extractors, neural networks will perform better than a random features linear predictor (NTK at init). It is fair to compare the after-kernel with finite-width NN as done in the previous work. The thing that I find unsatisfactory is why should one use the after-kernel as a linear predictor with learned features, say instead of re-training the last layer of the neural network. I think the more interesting part of the paper is the evolution of the ENTK.
>
> One reason to use the after-kernel instead of the final layer is capacity. The final layer (for the base network)  has 512 neurons which means that it’s performance will saturate after seeing c*512 samples for some small constant c. This does not apply to the after kernel, as we see in Figure 4c.
>
> >Overall I would appreciate more interpretations in terms of feature learning and also in general.
>
> One interpretation from Section 4 and 5 is that feature learning is not limited or concentrated to the initial part of training (as hypothesized by previous works, see Hypothesis 5.1). We also agree that exploring feature learning is an interesting direction but we felt that other explorations would be out of scope for this work.
>
> >I might be missing something but in Figure 5-a, the caption writes empirical NTK at initialization. How is it possible that it evolves during training if it is fixed at initialization? I am guessing that it improves because its training dataset also has grown. Is that correct?
> The text needs update accordingly, it is confusing as it is.
>
> By empirical NTK at initialization we mean the model obtained by constructing the empirical NTK of the neural network at initial weights. Let us call this kernel/model M.
>
> In Figure 5a the x-axis is the number of steps of training. In the bottom part the blue line represents the test error when we train M, so at x-axis value of t it represents the test error of M when we have trained it for t steps.
>
> We will update the text to be easier to read.
>
>
> >The more interesting insight would be what the scaling laws of NTKs tell us about the performance of neural networks. Are they helpful to predict or understand the generalization of finite-width neural networks?
> >Follow-up question: Is there any link between the performance of the after-kernel and the evolution speed of ENTK?
>
> While we do not study these questions, prior works suggest that when comparing architectures such as MLPs to CNNs all of these are correlated.
> Our intuition (due to the differences in scaling laws) is that while for smaller datasets scaling laws for NTKs might give us an indication of the performance of finite-width neural networks at larger scale the difference between them would be too large to conclude anything interesting. This approach is also limited by the fact that training NTKs on large datasets is compute prohibitive.
>
> >Typos
>
> Thank you for pointing out the typos, we will make sure to correct them.

---

### Review · Reviewer_Lg8z · 2023-04-24

**Summary Of Contributions:**

This paper reports several pieces of empirical evidence that illustrate the limitations of the neural tangent kernels---both the infinite width limit and the empirical NTK---in capturing and representing the behavior of neural networks. The focus is on the scaling behavior of neural networks and kernels with respect to the size of the dataset. The main finding is that NTKs have a noticeably worse scaling property than NNs, consistently over varying widths and hyperparameters for the NNs. In addition, the paper reports that the after-kernel continues to improve with the size of the dataset, and no noticeable "drift" during the early phase of training.

**Audience:**

Yes

**Broader Impact Concerns:**

I do not think this paper needs to address any ethical concerns.

**Claims And Evidence:**

Yes

**Requested Changes:**

Please address the "weaknesses" above.
- The experiments on other models will be too much for the given timeline, but I expect at least more justifications on why looking at small CNNs is enough, in authors' opinion.
- Also fix some typos: (1) Figure 2 caption, "widths and its the infinite NTK" -> "widths and the infinite NTK" (2) page 7, "kenel" -> "kernel." (3) Appendix B, what is "V Relu"?

**Strengths And Weaknesses:**

__Strengths.__
- The experiments are well-designed and conducted with a significant computational effort.
- The empirical results demonstrate a very clear pattern, making the claim very convincing (although this is usually true for many log-log style plots).
- The paper is well-organized and clearly written.

__Weaknesses.__
- The paper has a relatively weak followthrough; having observed the clear discrepancy between the scaling behaviors of NTK and NN, the paper does not attempt to provide an original explanation of why such a phenomenon takes place (or come up with an idea to take advantage of it). Without such followthrough, it is quite difficult to see the immediate merits of the observations over the prior works on the limitations of the NTK. Given the present manuscript, I can only think of one intellectual merit: a sanity check for the future theoretical framework that they should be able to explain this discrepancy in the scaling behavior. Other than that, I am not sure. In this regard, the paper should have given more detail on (1) in which regard the classical theory fails, and (2) in which sense extending the current theory to explain the observation is challenging and important.


- If I understood correctly, the experiments took place only on Myrtle CNN and 5-layer CNN, which are small and simplified versions of ConvNets. In this regard, I wonder whether the same pattern will also take place on other architectures without being confined to ConvNets.

- Some of the claims should be rephrased/corrected to avoid misunderstanding. On page 9, the paper argues that "we will give evidence that suggests, contrary to prior work, that there is no such "phase transition."" I believe that the correct way to put it should be "we did not observe any phase transitions under the experimental settings tested. In other words, the empirical results can only rigorously disprove the statement that "the phase transition __always__ takes place," and not proving that "the phase transition __never__ happens." Because this paper is almost 100% about experiments & observations, one must describe them very carefully. (In fact, I am not really sure if this claim is even true; the log-log scale does not seem to be an ideal way to capture such a phenomenon.)

- One could make this paper clearer to unexperienced readers (like me) by adding some detailed explanations and experimental setups. For instance, authors could clarify which initialization schemes the models use, and give more formal details on what "NTK parameterization" is.

---

> ### Author Response · Authors · 2023-05-03
> **Response to Reviewer Lg8z**
>
> >The paper has a relatively weak followthrough; having observed the clear discrepancy between the scaling behaviors of NTK and NN, the paper does not attempt to provide an original explanation of why such a phenomenon takes place (or come up with an idea to take advantage of it). Without such followthrough, it is quite difficult to see the immediate merits of the observations over the prior works on the limitations of the NTK. Given the present manuscript, I can only think of one intellectual merit: a sanity check for the future theoretical framework that they should be able to explain this discrepancy in the scaling behavior. Other than that, I am not sure. In this regard, the paper should have given more detail on (1) in which regard the classical theory fails, and (2) in which sense extending the current theory to explain the observation is challenging and important.
>
> The aim of the paper is not to explain the difference in scaling theoretically. The evolution of empirical NTK is an empirical explanation of the difference in scaling laws. Further, our study of how the NTK evolves (i.e. feature learning) due to training (Section 4 and 5) is in contrast to claims in prior works and presents an alternative view of feature learning where it is not concentrated on the initial part of training.
>
> Re (1), the NTK theory fails in our experiments because of the evolution of empirical NTK. Experimentally, the final empirical NTK is sufficient to explain the performance of the final network (as shown by prior works and also our experiments).
>
>
> >If I understood correctly, the experiments took place only on Myrtle CNN and 5-layer CNN, which are small and simplified versions of ConvNets. In this regard, I wonder whether the same pattern will also take place on other architectures without being confined to ConvNets.
> >The experiments on other models will be too much for the given timeline, but I expect at least more justifications on why looking at small CNNs is enough, in authors' opinion.
>
>
> We agree that this is an interesting future direction. We believe this pattern will hold even more strongly for networks such as vision transformers which lack* inductive bias and rely more heavily on feature learning facilitated by large datasets. This can also be seen in the observation that Vision Transformers are only competitive or better than CNNs in the large data regime. Since there is no feature learning in a kernel this benefit will not apply to corresponding NTKs.
>
>
> *as compared to CNNs for vision tasks.
>
> >Some of the claims should be rephrased/corrected to avoid misunderstanding. On page 9, the paper argues that "we will give evidence that suggests, contrary to prior work, that there is no such "phase transition."" I believe that the correct way to put it should be "we did not observe any phase transitions under the experimental settings tested. In other words, the empirical results can only rigorously disprove the statement that "the phase transition always takes place," and not proving that "the phase transition never happens." Because this paper is almost 100% about experiments & observations, one must describe them very carefully. (In fact, I am not really sure if this claim is even true; the log-log scale does not seem to be an ideal way to capture such a phenomenon.)
>
> We agree and we will make this change.
>
> Wrt “the log-log scale does not seem to be an ideal way to capture such a phenomenon”: We think that the log-log scale is the right scale since this is the scale at which we see scaling laws. If we plot loss curves instead of a normal scale we will also observe that most of the improvement in loss comes early in training, clearly this is not considered  to be a phase transition.
>
> >One could make this paper clearer to unexperienced readers (like me) by adding some detailed explanations and experimental setups. For instance, authors could clarify which initialization schemes the models use, and give more formal details on what "NTK parameterization" is.
> >Also fix some typos:..
>
> Thanks for pointing these out, we will make these changes. V in "V Relu" is a typo, it is just a Relu.

---

### Decision · Action_Editors · 2023-06-27

**Recommendation:** Accept with minor revision

**Comment:**

### Summmary
The paper examines how the neural tangent kernel (NTK) scales with dataset size. The main finding is an empirical observation that the NTK scales much worse than neural networks (NN), highlighting a fundamental difference between NN and NTK. This observation holds true for both infinite-width and empirical NTKs. Additionally, the paper demonstrates that empirical NTK evolves continuously during training, contrary to the previous belief that it stabilizes after a few epochs. These results emphasize the limitations of the current NTK approach for studying the scaling law of NN.

### Reviewer Recommendations
The reviewers generally agree that the empirical observations are interesting, and it's a good fit for TMLR. However, their main concerns are on the overclaim in the contribution and title. Specifically, the reviewers pointed out that the paper should clarify that its results are "empirical" and limited to "scaling laws," especially in the title.

### AE Recommendation
The AE agrees with the reviewers' concern. Therefore, my recommendation is to accept the paper with minor revisions. We expect the authors to clarify their contribution in the revision. This includes:
- Change the title to "Empirical Limitations of the NTK for Understanding Scaling Laws in Deep Learning"
- Clarify in their abstract and main text that their contributions focus on "empirical" observations and "scaling laws" rather than generalization.

**Audience:**

The paper aligns with the interests of the TMLR audience.

**Claims And Evidence:**

The results are generally accurate, supported by empirical evidences. However, the claims could be toned down. For example, it would be beneficial to clarify that the scope is limited to "scaling laws" rather than "generalization." See the detailed comments.